# Routine and Advanced Laboratory Tests for Hemostasis Disorders in COVID-19 Patients: A Prospective Cohort Study

**DOI:** 10.3390/jcm11051383

**Published:** 2022-03-03

**Authors:** Paul Billoir, Perrine Leprêtre, Caroline Thill, Jeremy Bellien, Veronique Le Cam Duchez, Jean Selim, Fabienne Tamion, Thomas Clavier, Emmanuel Besnier

**Affiliations:** 1Vascular Hemostasis Unit, CHU Rouen, Normandie University, UNIROUEN, INSERM U1096, F-76000 Rouen, France; paul.billoir@chu-rouen.fr (P.B.); veronique.le-cam@chu-rouen.fr (V.L.C.D.); 2Medical Intensive Care Unit, CHU Rouen, Normandie University, UNIROUEN, INSERM U1096, F-76000 Rouen, France; perrine.lepretre@chu-rouen.fr (P.L.); fabienne.tamion@chu-rouen.fr (F.T.); 3Department of Biostatistics, CHU Rouen, Normandie University, UNIROUEN, INSERM 1404, F-76000 Rouen, France; caroline.thill@chu-rouen.fr; 4Department of Pharmacology, CHU Rouen, Normandie University, UNIROUEN, INSERM U1096, F-76000 Rouen, France; jeremy.bellien@chu-rouen.fr; 5Department of Anesthesiology and Critical Care, CHU Rouen, Normandie University, UNIROUEN, INSERM U1096, F-76000 Rouen, France; jean.selim@chu-rouen.fr (J.S.); thomas.clavier@chu-rouen.fr (T.C.); 6Centre d’Investigation Clinique, CHU Rouen, F-76000 Rouen, France

**Keywords:** SARS-CoV-2, COVID-19, intensive care unit, critical care outcomes, hemostasis, platelet

## Abstract

Background: Thrombosis is frequent during COVID-19 disease, and thus, identifying predictive factors of hemostasis associated with a poor prognosis is of interest. The objective was to explore coagulation disorders as early predictors of worsening critical conditions in the intensive care unit (ICU) using routine and more advanced explorations. Materials: Blood samples within 24 h of ICU admission for viscoelastic point-of-care testing, (VET), advanced laboratory tests: absolute immature platelet count (A-IPC), von Willebrand-GPIb activity (vWF-GpIb), prothrombin fragments 1 + 2 (F1 + 2), and the thrombin generation assay (TGA) were used. An association with worse outcomes was explored using univariable and multivariable analyses. Worsening was defined as death or the need for organ support. Results: An amount of 85 patients with 33 in critical condition were included. A-IPC were lower in worsening patients (9.6 [6.4–12.5] vs. 12.3 [8.3–20.7], *p* = 0.02) while fibrinogen (6.9 [6.1–7.7] vs. 6.2 [5.4–6.9], *p* = 0.03), vWF-GpIb (286 [265–389] vs. 268 [216–326], *p* = 0.03) and F1 + 2 (226 [151–578] vs. 155 [129–248], *p* = 0.01) were higher. There was no difference observed for D-dimer, TGA or VET. SAPS-II and A-IPC were independently associated with worsening (OR = 1.11 [1.06–1.17] and OR = 0.47 [0.25–0.76] respectively). The association of a SAPS-II ≥ 33 and an A-IPC ≤ 12.6 G/L predicted the worsening of patients (sensitivity 58%, specificity 89%). Conclusions: Immature platelets are early predictors of worsening in severe COVID-19 patients, suggesting a key role of thrombopoiesis in the adaption of an organism to SARS-CoV-2 infection.

## 1. Introduction

Coronavirus disease 2019 (COVID-19) is a pandemic with major effects on morbidity and mortality, responsible for millions of deaths worldwide [1,2].

Alteration in coagulation processes is frequently observed, which can favor thrombo-embolic events and pulmonary micro-thrombi, worsening hypoxemia [3]. Coagulation over-activation induces tissue factor synthesis, which activates factor VII, generating factor Xa, thrombin and fibrin, and ultimately induces a pro-coagulant state in the microvasculature of many critical organs, participating in organ failure and death [4,5,6]. This hyperactivation of coagulation is not easily detected with routine laboratory tests [7]. Nevertheless, other approaches may help to identify patients with hypercoagulation profiles. Some studies have observed a shortening of clotting time, high fibrinogen activity and fibrinolysis shutdown using viscoelastic tests (VET) [8,9,10,11]. More advanced laboratory tests may also be of interest. Prothrombin fragments 1 + 2 (F1 + 2) are markers of thrombin generation, and therefore hypercoagulability. They have been identified as associated with thrombosis in COVID-19 patients [12]. Immature platelets (IP) are other parameters of interest. They reflect the potential of bone marrow for the production of platelets, i.e., thrombopoiesis [13]. Its count is an automated measure of reticulated platelets in peripheral blood. Relative proportions of IP have been described as early predictors of sepsis in ICU patients, and more recently, higher values of relative or absolute immature platelet count (A-IPC) have been observed in COVID-19 patients [14,15]. Finally, thrombin generation assay (TGA) is a technique that studies the initiation, propagation and inhibition of coagulation, allowing the observation of hypo- or hyper-coagulable risk profiles [16,17]. These three assays (F1 + 2, A-IPC and TGA) have been recently shown to be associated with the severity of COVID-19 patients and to a higher risk of intensive care unit (ICU) admission [15,18]. However, no study has explored the association between hematological and hemostasis markers (including A-IPC, F1 + 2 and TGA) and the worsening in organ failure during an ICU stay, nor the role of these markers as early predictors of such a worsening.

The aim of this study was to investigate the association of coagulation and blood count disorders as early predictors of the worsening of severe COVID-19 patients using traditional laboratory tests and more advanced explorations, including a “bedside” approach based on VET.

## 2. Materials and Methods

### 2.1. Study Design and Patients

We conducted a prospective monocenter cohort study in three intensive care units of a tertiary university hospital (a medical ICU, a surgical ICU and an “ephemeral” COVIDICU). These ICUs were familiar with the care of COVID-19 patients since the first epidemic wave in spring 2020. This study was approved by an ethics committee on 7 April 2020 under the approval number 2020-A00885-34 (Comité de Protection des Personnes Est-III) [19]. The study was performed in accordance with the principles of the Declaration of Helsinki related to human research. Because of the non-interventional nature of the study, the ethics committee waived the need for written consent, and patients were included after obtaining a verbal agreement to participate. The elaboration of the manuscript was in accordance with the STROBE statement. This study was registered before the first inclusion in the clinicaltrial.gov database (NCT04357847).

Patients were eligible for inclusion if they presented with a “severe” condition related to a documented infection to SARS-CoV-2 (by polymerase chain reaction assay), defined as the need for ICU admission because of an acute respiratory failure. Non-inclusion criteria were pregnancy, a documented bacterial co-infection, known limitations in life-support because of patient choice or important comorbidities, and an expected death within 24 h.

Patients were followed up to 28 days or death.

### 2.2. Endpoints

The primary endpoint was the worsening of clinical status from a “severe” condition towards a “critical” condition within 28 days after ICU admission, according to the NMA-Covid Initiative Network [20]. This definition is elaborated on in the NIH COVID-19 treatment guidelines [21]. In this definition, a “severe” condition is defined as the need for non-invasive ventilation, whereas a “critical” condition is defined as at least one of the following outcomes: (1) need for invasive mechanical ventilation, (2) shock (i.e., vasopressors need), (3) other organ failures (4) death.

The secondary endpoints were the abnormalities of coagulation tests or blood count described below, and their association with a worse outcome.

### 2.3. Blood Samples

Whole blood was collected on ethylenediaminetetraacetic acid (EDTA) and citrated tubes within the first 24 h after admission in the ICU, either using a venous or arterial catheter according to the available route. There was no heparin added to the tubing system of these routes. Platelet-poor plasma was prepared by double centrifugation of a citrated tube, for 15 min at 2250× *g* at room temperature, and frozen at −80 °C within 4 h after collection. Clinical and biologically relevant data were collected.

### 2.4. Assays

Blood count, routine biochemistry (creatinine, ionogram, liver enzymes, troponin and NT-pro-BNP) and coagulation tests were performed according to usual care at the reception of tubes. A complete blood count was performed on EDTA samples on the XN-1000 (Sysmex, Villepinte, France). The absolute immature platelet count (A-IPC) was determined with a fluorescent method. This method improves the gating of platelets, using side fluorescence (reflecting RNA content), a side scatter (intracellular structure) and forward scatter (cell sizer).

For coagulation, prothrombin time (PT), activated partial thrombin time (aPTT) (Diagnostica Stago, Snières-sur-Seine, France) and D-dimer (Vidas Dex2, Biomérieux, Marcy l’étoile, France) assays were performed according to usual care in our units within one hour after delivery to the laboratory.

After defrosting, several specific coagulation tests were assayed:
Fibrinogen (STA-Liquid Fib, Diagnostica Stago, Asnières-sur-Seine, France)Von Willebrand activity was evaluated with vWF:GPIb-binding activity (InnovanceVWAc, Siemens Healthcare, Marburg, Germany), assayed on the BCS XP (Siemens Healthcare, Marburg, Germany). The assay evaluated the capacity of the von Willebrand factor to bind glycoprotein Ib. The agglutination of coated bead was quantified by turbidimetry and expressed as UI/dL.Prothrombin fragments 1 + 2 were assayed with an enzyme-linked immunosorbent assay (Enzygnost F1 + 2, Siemens Healthcare, Marburg, Germany) using a Diasorin EtiMax.Thrombin generation assay (TGA) was triggered by a PPP reagent (5 pM of tissue factor) (Diagnostica Stago, Asnières-sur-Seine, France). TGA was measured by calibrated automated thrombography and the Fluorocan Ascent Fluorometer (Thermo Scientific Lab Systems, Helsinki, Finland). Five parameters were evaluated with TGA: the time to observe the first thrombin trace (lag time), the time to have the maximal thrombin generated (time to peak (TTP)), the maximal thrombin generation (thrombin peak) and the speed at which it forms (velocity). In a second time, a physiological inhibitor of coagulation was involved, and the global thrombin formation was calculated (endogenous thrombin potential (ETP)).


### 2.5. Viscoelastic Test

A blood sample on a citrated tube was analyzed through a point-of-care viscoelastic test (VET) using the Quantra^®^ device with Qplus cartridges^®^ (Hemosonics, Charlottesville, VI, USA; Stago, Hong Kong, China). Briefly, this device uses sonorheometry to explore ex-vivo the formation and the stiffness of the blood clot with four different channels: channel one contains kaolin to activate clot formation and study the coagulation time (CT); channel two contains kaolin + heparinase and thus permits to explore the coagulation time after the neutralization of the circulating heparin (CTH); channel 3 contains thromboplastin and thus explores the clot stiffness (CS) expressed in hPa as a result of fibrin polymerization and platelet adhesion; channel 4 contains thromboplastin + abciximab, neutralizing platelets’ adhesion, and thus exploring the specific participation of fibrinogen to clot stiffness (FCS in hPa). The platelet contribution to clot stiffness (PCS in hPa) = CS − FCS. Because most patients benefited from anticoagulant therapy, we only considered CTH for clotting time.

Because VET has been developed for the monitoring of bleeding in perioperative settings and not for hypercoagulant states, we compared data with those from 25 pre-operative cardiac surgery patients, free of anticoagulant therapies or hemostasis pathologies, and thus, considered as “control” patients.

The reference values of the different laboratory tests used in this study are presented in the Appendix A.

### 2.6. Data and Statistical Analysis

The primary objective of the study was to evaluate the association of blood count, hemostasis and clinical characteristics on admission in the ICU with the occurrence of a worsening towards “critical” condition during the first 28 days in ICU, as previously defined in Section 2.2. Severe patients who did not fill the “critical” criteria were defined as “non-critical”

Because of the non-normal distribution of most of the data, as observed using a D’Agostino test, results are expressed as medians with interquartile ranges (IQR) for quantitative data, and as absolute numbers and percentages (n, %) for qualitative data.

First, comparisons between “critical” and “non-critical” groups were performed. Because of the non-normal distribution of most of the quantitative data, comparisons were performed using a Mann–Whitney two-tailed test for unpaired values. A Fisher’s exact test was used for qualitative data. A *p*-value < 0.05 was considered statistically significant.

Then, we assessed the association between worsening into a critical condition and the characteristics of interest at the ICU admission as early predictors. A univariable logistic regression analysis of clinical outcome (“non-critical” or “critical”) was performed using the following variables at ICU admission as predictors: age, body mass index (BMI), Simplified Acute Physiology Score II (SAPS-II), treatment with a corticosteroid or anticoagulant at admission, and biological markers of interest: neutrophils to lymphocytes ratio (NLR), neutrophils to platelets ratio (NPR), immature platelets, D-dimer, fibrinogen, vWF:GPIb-binding activity and prothrombin F1 + 2. Variables presenting a *v* ≤ 0.1 in this univariable analysis were included in a multivariable analysis using a logistic regression model with a backward stepwise process, to adjust for confounders (logistic regression full model). Furthermore, *p*-values were computed using Wald’s method. A Hosmer–Lemeshow test was computed to evaluate the fitness of logistic regression with presented data. To confirm the independence of variables and provide a more precise odds ratio (OR), we performed a second logistic regression, including only variables with a significant association in the initial full model (logistic regression final model). Thereafter, receiver operating characteristic (ROC) curves for clinically significant factors identified in the multivariable analysis were constructed, and the area under the curve (AUC) was calculated with a 95% confidence interval. Optimal cut-off values, defined as the best combination of sensitivity and specificity, were determined using the Youden index.

All analyses were performed with GraphPad Prism v9.1.2 for Windows (Graphpad, San Diego, CA, USA).

## 3. Results

One hundred patients were included between May and October 2020. Among them, two were excluded because of a wrong diagnosis of COVID-19 pneumonitis (a patient with a respiratory failure secondarily attributable to interstitial lung disease, and another with a false positivity of PCR), and thirteen patients because of technical issues during laboratory experiments (five because of early coagulation within the sample tube and eight because of an absence of clotting throughout the analyses, most likely due to technical issues). Thus, 85 patients were analyzed and followed up to their hospital discharge or death. Clinical and biological characteristics of patients are presented in Table 1 and outcomes are given in Table 2.

Briefly, 33 patients were considered as clinically worsening into a critical condition during their ICU stay. A total of 31 out of 33 required invasive mechanical ventilation, and among them, 24 needed a norepinephrine infusion, 1 required a renal replacement therapy and 8 died. All patients completed the follow-up. Age, SAPS-II and SOFA severity scores at ICU admission were significantly different between “critical” and “non-critical” groups. Notably, higher SOFA scores were observed up to 7 days after ICU admission, a longer duration of respiratory support and of ICU stay. As expected, the duration of invasive mechanical ventilation and ICU length of stay was longer in the “critical” group. Critical patients presented a higher rate of renal failure, with a higher proportion of KDIGO score ≥ 1. Finally, critical patients presented higher plasma levels of urea, troponin and NT-pro-BNP at admission. No difference was observed for chronic underlying diseases and for treatments at admission. The vast majority of patients were treated with corticosteroids and anticoagulant therapies at ICU admission, with no difference between groups. Anticoagulation was performed using low-molecular-weight-heparin at doses following the guidelines from the French Society of Anesthesia and Intensive Care.

Biological characteristics of interest at admission are presented in Table 3 and Figure 1.

Concerning blood counts, we observed a significant difference between groups for A-IPC, lymphocytes, NLR and NPR. Concerning hemostasis biomarkers, fibrinogen levels, vWF:GPIb-binding activity and F1 + 2 were also increased in the critical group. Thrombin generation assays showed no difference between groups (Figure 2 and see Appendix A for the descriptive curves of thrombin generation).

VET profiles showed a pro-coagulant state in COVID-19 patients versus “control” patients, with shorter clotting times and stronger clot stiffness, fibrinogen and platelet contributions. No difference was observed between “critical and non-critical groups (Table 4).

### 3.1. Association between Clinical-Biological Parameters and Clinical Outcome

Results for the univariable and multivariable analyses are presented in Table 5. Among univariable analyses, age, SAPS-II, NLR, vWF activity and F 1 + 2 presented a significant association with worsening. Concerning the multivariable analysis, only SAPS-II (OR = 1.1 [1.06–1.17]) and A-IPC count (OR = 0.47 [0.25–0.76]) were independently associated with the worsening of patients towards a critical condition within the first 28 days in ICU. The Hosmer–Lemeshow test presented a *p*-value of 0.6, assuming the logistic regression model satisfactorily fitted with data.

### 3.2. ROC Curves Analyses

ROC curves were elaborated for SAPS-II and A-IPC for the prediction of worsening (Figure 3). The ROC curve for SAPS-II presented an AUC of 0.804 [0.704–0.882] with a cut-off value ≥ 33 for the prediction of worsening, with a sensitivity of 78.8% and a specificity of 75.0%. The ROC curve for A-IPC presented an AUC of 0.660 [0.547–0.762] with a cut-off value ≤ 12.6 G/L for prediction, with a sensitivity of 81.2% and a specificity of 51.0%. The AUC of these two ROC curves was not statistically different for the prediction of worsening (*p* = 0.13). Nonetheless, we can estimate that the combination of a SAPS-II ≥ 33 and an A-IPC ≤ 12.6 G/L presented a specificity of 89% and a sensitivity of 58% to predict the worsening of patients (Figure 4).

## 4. Discussion

We provided evidence that patients experiencing a worsening of their clinical condition in ICU presented several alterations in the coagulation process (fibrinogen, F 1 + 2 and vWF activity). However, the most important result of our study concerns the association between IP and worsening. IP fractions or counts have been established to measure bone marrow platelet production, as reticulocyte count for red cells [22]. They are the youngest circulating platelets, derived from megakaryocytes, and thus present a larger size, an important RNA content, and a higher activity [23]. Thus, they are suspected to be implicated in thrombosis during some pathological settings. Indeed, higher troponin levels were associated with higher IP fractions during acute myocardial injury, with a subsequent association with worse cardiac outcomes [24]. During sepsis, higher A-IPC seems to be associated with the severity and prognosis of patients [25]. These high levels of IP may be related to the intense inflammatory process and/or take part in the pathophysiological process leading to disseminated intravascular coagulation, which may explain some of the observed associations with prognosis [26]. Nevertheless, another cohort study observed an association between lower A-IPC and the occurrence of severe thrombocytopenia, but also with mortality [27], suggesting on the opposite, that a defect in thrombopoiesis contributes to poor prognosis during sepsis. Few studies have explored A-IPC during COVID-19 disease. A recent study in 658 COVID-19 patients observed that higher A-IPC were associated with a longer length of stay and with the need for ICU admission [15]. This study observed an association between death or ventilator use with either the peak A-IPC during hospitalization or A-IPC at hospital admission. Thus, the timing of blood sampling differs from our cohort, for which we focused on A-IPC at admission in ICU when the pathophysiological process related to COVID-19 gets worse. Moreover, this study included not only patients in ICU but also in the general ward [15]. It is therefore difficult to determine if the increase in A-IPC is an adaptative process to the severity of the infection or a deregulation in thrombopoiesis. In another retrospective study, some authors observed in 47 COVID-19 patients that the IP count was higher than that of stable patients with cardiovascular risk factors but similar to that of patients admitted for acute myocardial infarction [28]. We also observed similar high A-IPC in our study, with a median value of around 10 G/L in our entire cohort. Nevertheless, and contrary to the two previous studies, we observed a relatively lower A-IPC in “critical” patients, and the multivariable analysis allowed us to show the association of lower A-IPC with clinical outcomes. As discussed above, for sepsis, the pathophysiological significance of these conflicting results is difficult to interpret. On one hand, SARS-CoV-2-related inflammation may enhance thrombopoiesis, notably through a cytokine-induced production of thrombopoietin [29], explaining the global high levels of A-IPC in the various studies, including ours. On the other hand, some observations described the occurrence of thrombocytopenia during ICU stays of the most severe COVID-19 patients, with an incidence of up to 30% and a significant association with mortality [30,31,32]. Several hypotheses have been raised to explain these observations: increased platelet destruction and/or consumption, secondary hemophagocytic lymphohistiocytosis and a reduction in production. This latter hypothesis is supported by the ability of SARS-CoV-2 to infect bone marrow cells and, thus, possibly megakaryocytes [32]. Our findings may reinforce the hypothesis of bone marrow consideration in the most serious COVID-19 cases, resulting in the observed relative reduction in AIP-C. This observation of an increased physiological process, but to a lesser extent in the most severe patients, has also been described in other systems during sepsis. This is notably the case of the relative adrenal insufficiency described in the most severe cases of sepsis [33]. Thus, we can raise the hypothesis, by analogy, of a relative thrombopoietic insufficiency related to severe infections, including those of SARS-CoV-2. Because we chose to analyze samples at an early and given time for all patients, we described the usefulness of A-IPC as an early predictor of worsening in patients with severe conditions.

Concerning the coagulation biomarkers, Zhang et al. suggested that an increase of D-dimer is associated with poor prognosis [34]. Nevertheless, and similarly to our cohort, other studies failed to confirm this result [7,35]. Thus, it seems that the D-dimer is an unreliable marker of unfavorable evolution in severe COVID-19 patients. We showed VET prothrombotic profiles in COVID-19 patients as previously described for all current devices, including Quantra^®^ [8,36,37,38]. If most of the studies demonstrated higher disturbances in the most severe patients, no study has explored the independent association of VET with prognosis. Because we found no difference between severe and critical patients in our cohort, we can suggest that VET lacks a sensitivity to discriminate patients with worse outcomes. Nevertheless, the Quantra Qplus cartridges used in our study, do not explore fibrinolysis, whereas an important shutdown of fibrinolysis has been observed during COVID-19 disease and predicted thrombotic events [38]. Thus, results may have been different with the Qstat cartridges dedicated to fibrinolysis exploration. Nevertheless, these cartridges do not contain heparanase, whereas almost all patients were under heparin therapy, and thus, this would not have permitted us to explore intrinsic hemostasis alteration.

Concerning other markers, F1 + 2 were increased in worse patients. F1 + 2 are generated during the proteolytic formation of thrombin and are therefore early markers of coagulation. In a cohort study of SARS-CoV-2-related ARDS, F1 + 2 plasma levels at ICU admission were higher in non-survivors [39], and in another study, values > 500 pM were associated with venous thromboembolism [12]. Our study showed a three-fold proportion of patients presenting F1 + 2 values > 500 pM in the “worsening” group. We recently described an association between increased levels of fibrinogen, thrombin generation and F1 + 2 and the need for ICU admission in patients hospitalized for SARS-CoV-2 pneumonitis in a general ward [18]. However, the results from our present study do not suggest such an association. The major discrepancy between the present study and our previous one is the greater proportion of patients under anticoagulant therapy at ICU admission, which may have prevented some of the thrombin generations. This can also explain the absence of difference for TGA assays.

Endothelial lesions and coagulation disorders are closely linked. Because the vWF factor is stored in endothelial cells, the exploration of its activity may reflect the activation of the endothelium in pathological settings. Its release from endothelial cells occurs during inflammatory processes and ultimately deregulated hemostatic processes may occur, notably because of ADAMTS 13’s unbalanced function, at least partially explaining the thrombotic state [40,41]. Thus, vWF activity may predict mortality during systemic inflammatory syndrome [42], including COVID-19 worse situations [43]. In our study, the vWF activity was increased in “critical” patients, but no independent association with worse outcomes was observed.

When assessing the clinical evolution of COVID-19 patients in ICU according to SAPS-II and A-IPC as predictors of worsening, we determined cut-off values of 33 for SAPS-II and 12.6 G/L for A-IPC. SAPS-II presented a good AUC and good sensitivity and specificity, whereas A-IPC presented a lower AUC, a higher sensitivity but a poor specificity. Globally, SAPS-II appeared more effective in predicting worsening in comparison with A-IPC. The combination of SAPS-II and A-IPC presented a higher specificity for the prediction of worsening in comparison with the sole parameter, which may help physicians to identify patients and thus may optimize the triage of patients between ICU and intermediate care units.

Our study presents several limitations. We have a limited sample size and this may explain the absence of significance for some markers, notably F1 + 2 and vWF activity, whereas significant differences between groups were observed. Nevertheless, this absence of association may also be due to the intrinsic association between all these hemostatic mechanisms, limiting the independence in the logistic regression model. Our study may be limited by its monocentric enrollment. Another limitation was the technical measurement of A-IPC. The use of fluorescent dyes to stain immature platelets carries a tendency of these dyes to non-specifically bind to other platelet components in addition to RNA [13]. However, the Sysmex XN analyzer used a dye very specific to platelet RNA.

## 5. Conclusions

Our study suggests that the alteration in adaptative thrombopoiesis, reflected by an underproduction of IP, may serve as an easily accessible biomarker to predict outcomes among critically ill COVID-19 patients. Further studies are needed to consider A-IPC measurements in the clinical assessment of these patients.

## Figures and Tables

**Figure 1 jcm-11-01383-f001:**
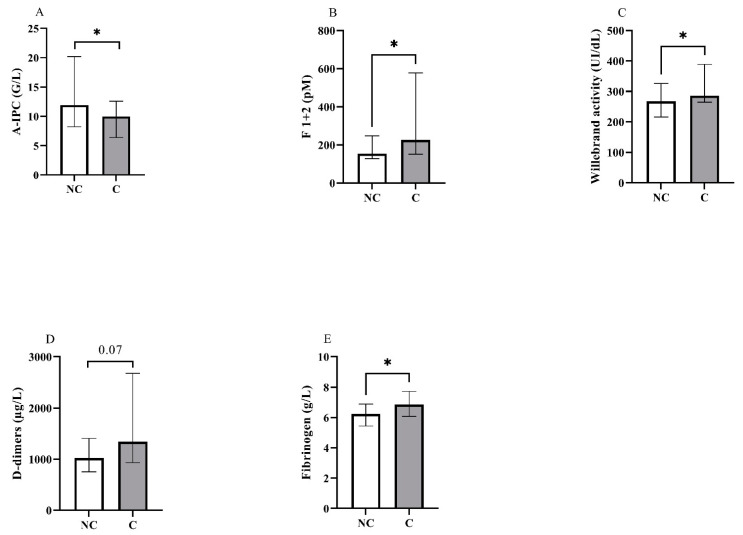
Representation in non-critical (NC) and critical (C) groups of the blood levels for (**A**) absolute immature platelet count (A-IPC), (**B**) prothrombin fragments 1 + 2, (**C**) von Willebrand activity, (**D**) D-dimers and (**E**) fibrinogen. *: *p* < 0.05.

**Figure 2 jcm-11-01383-f002:**
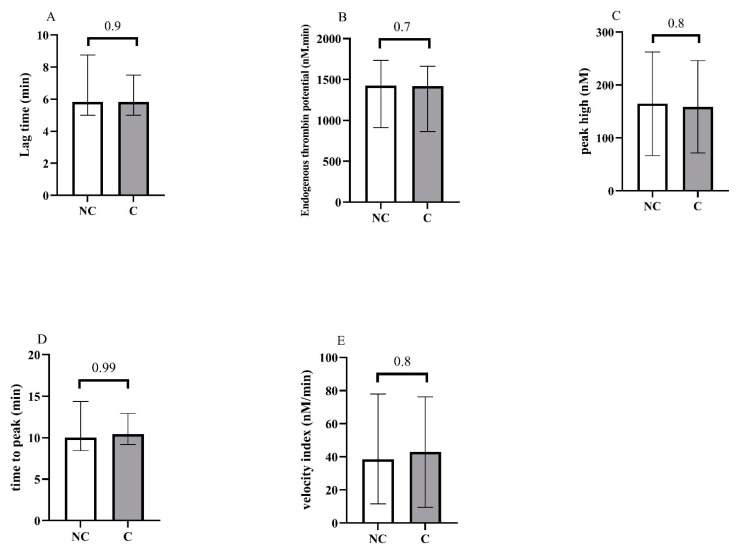
Representation of the thrombin generation assays at the admission of patients in the ICU, and according to non-critical (NC) and critical (C) groups. (**A**) Lag time, (**B**) endogenous thrombin potential, (**C**) peak high, (**D**) time to peak, (**E**) velocity.

**Figure 3 jcm-11-01383-f003:**
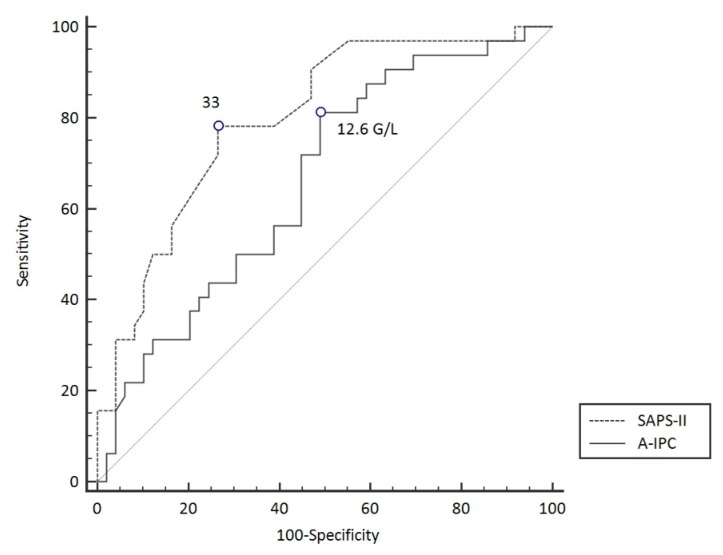
ROC curve models to predict worsening in COVID-19 patients.

**Figure 4 jcm-11-01383-f004:**
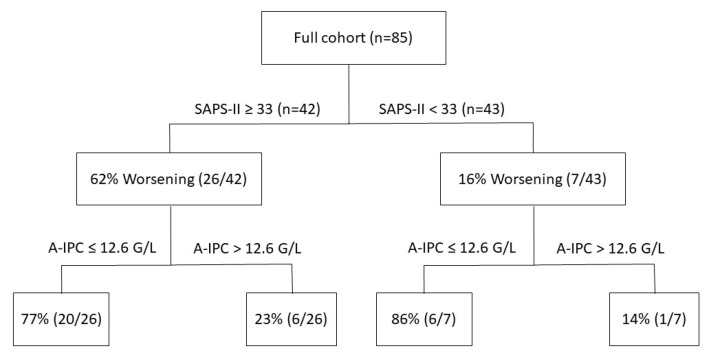
Repartition of patients according to SAPS-II severity score values and absolute immature platelet counts and their respective evolution during their ICU stay (worsening or not worsening).

**Table 1 jcm-11-01383-t001:** Clinical and biological characteristics at admission in ICU.

Parameters	All (n = 85)	Non-Critical (n = 52)	Critical (n = 33)	*p*-Value
Age (years)	67 [58.0–72.5]	60.5 [53.3–70.8]	70 [64.0–74.5]	<0.001
Male	58 (68.2%)	35 (67.3)	23 (69.7%)	>0.99
BMI (kg/m^2^)	28.8 [25.1–34.2]	29.3 [25.2–35.6]	28.1 [24.8–32.6]	0.3
Obesity (BMI ≥ 30 kg/m^2^) (n, %)	38 (44.7%)	24 (46.2%)	14 (42.4%)	0.8
Underlying comorbidity (n, %)				
Chronic obstructive pulmonary disease	10 (11.7%)	6 (11.5%)	4 (12.1%)	0.8
Asthma	13 (15.3%)	10 (19.2%)	3 (9.1%)	0.2
Diabetes	36 (42.4%)	20 (38.5%)	16 (48.5%)	0.4
Hypertension	50 (58.8%)	30 (57.7%)	20 (60.6%)	0.8
Peripheral arterial disease	2 (2.4%)	0 (0%)	2 (6.1%)	0.1
Coronaropathy	8 (9.4%)	4 (7.7%)	4 (12.1%)	0.7
Smoking history	2 (2.4%)	1 (1.9%)	1 (3.0%)	>0.99
Active neoplasia	6 (7.1%)	5 (9.6%)	1 (3.0%)	0.4
eGFR at admission (mL/min/m^2^)	66.0 [51.5–85.9]	63.5 [52.0–80.5]	72.5 [46.5–98.1]	0.4
COVID-19 related treatment at admission (n, %)				
Corticosteroid	81 (95.3%)	51 (98.1%)	30 (90.9%)	0.29
Remdesivir	14 (16.5%)	11 (21.2%)	3 (9.1%)	0.2
Lopinavir/ritonavir	6 (7.1%)	4 (7.7%)	2 (6.1%)	>0.99
Tocilizumab	0 (0%)	0 (0%)	0 (0%)	NA
Anticoagulation therapy (n, %)	81 (95.3%)	49 (96.2%)	29 (87.9%)	0.4
Prophylactic intensity	6 (7.1%)	3 (5.8%)	3 (9%)	0.7
Intermediate intensity	58 (68.2%)	40 (76.9%)	18 (54.5%)	0.055
Therapeutic intensity	14 (16.5%)	6 (11.5%)	8 (24.2%)	0.1
ICU transfer since the onset of symptoms (days)	9.0 [6.0–10.0]	9.0 [7.0–11.0]	8.0 [4.5–10.0]	0.043
SAPS-II score	32 [25–40]	27 [22–34.5]	40 [34–53.5]	<0.001
SOFA score	2 [1–4]	2 [1–3]	3 [2–5]	<0.001
Non-invasive respiratory support (HFNC or NIV) (n, %)	80 (94.1%)	51 (98.1%)	29 (87.9%]	0.07
PaO2/FiO2 ratio (n = 83)	136 [92–168]	149 [110–168]	115 [77–169]	0.06
Biological parameters				
Urea (mmol/L)	6.7 [4.7–9.1]	5.5 [4.5–7.9]	8.6 [5.9–12.2]	0.002
C Reactive Protein (n = 83) (mg/L)	122 [71–196]	122 [76–204]	131 [63–190]	0.8
High-sensitive troponin (n = 75) (ng/L)	13 [7–22]	10 [6–17]	16 [9–37]	0.004
NT-pro-BNP (n = 74) (ng/L)	199 [76–609]	169 [63–419]	261 [114–1061]	0.04
Lactatemia (mmol/L)	1.4 [1.0–1.7]	1.3 [0.9–1.7]	1.4 [1.1–1.7]	0.3

BMI: body mass index; eGFR: estimated glomerular filtration rate by the modification of diet in renal disease formula; HFNC: high flow nasal canula; ICU: intensive care unit; NIV: non-invasive ventilation; SAPS-II: Simplified Acute Physiology Score II; SOFA: sepsis-related organ failure assessment.

**Table 2 jcm-11-01383-t002:** Clinical outcomes during the ICU stay of patients admitted with COVID-19 pneumonitis.

Parameters	All (n = 85)	Non-Critical (n = 52)	Critical (n = 33)	*p*-Value
Prognostic score				
SOFA_D0_	2 [1–4]	2 [1–3]	3 [2–5]	<0.001
SOFA_D3_	2 [2–4]	2 [0–2]	6 [2–8]	<0.001
SOFA_D7_	1 [0–3]	0 [0–1]	4 [3–6]	<0.001
Invasive mechanical ventilation (n, %)	31 (36.5%]	0 (0%)	31 (93.9%)	NA
Duration of invasive ventilation (days)		0	12 [6–21]	NA
KDIGO score ≥ 1 (n, %)	43 (50.6%)	16 (30.8%)	27 (81.8%)	<0.001
Renal Replacement Therapy	1 (1,2%)	0 (0%)	1 (3%)	NA
Norepinephrine use (n, %)	24 (28.2%)	0 (0%)	24 (72.7%)	NA
Thrombosis	9 (10.6%)	3 (5.8%)	6 (18.2%)	0.051
Length of ICU stay (days)	9 [5–23]	6 [4–9]	27 [16–28]	<0.001
Mortality (n, %)	10 (11.8%)	0 (0%)	10 (30%)	NA

NA: Not Applicable. SOFA scores were calculated at admission (D0), day-3 (D3) and day-7 (D7).

**Table 3 jcm-11-01383-t003:** Blood count and hemostasis assays at admission in ICU.

Parameters	All (n = 85)	Non-Critical (n = 52)	Critical (n = 33)	*p-*Value
Hemoglobin (g/dL)	12.8 [11.7–13.9]	12.8 [11.7–13.9]	12.9 [10.1–13.8]	0.7
Reticulocytes (n = 78) (G/L)	30.1 [22.5–42.5]	31.7 [24.2–44.7]	28.1 [21.4–39.9]	0.1
Platelets (G/L)	283 [215–365]	300 [224–374]	264 [187–332]	0.1
A-IPC (n = 80) (G/L)	10.3 [7.6–15.5]	12.3 [8.3–20.7]	9.6 [6.4–12.5]	0.02
Leukocytes (G/L)	8.3 [6.1–11.5]	7.5 [6.0–10.6]	8.5 [6.1–12.1]	0.5
Neutrophils (G/L)	6.9 [4.7–9.6]	6.4 [4.7–8.8]	7.8 [5.1–11.0]	0.3
Lymphocytes (G/L)	0.62 [0.43–0.98]	0.67 [0.52–1.09]	0.54 [0.34–0.79]	0.01
Monocytes (G/L)	0.30 [0.19–0.43]	0.29 [0.17–0.43]	0.30 [0.21–0.46]	0.3
Neutrophils/lymphocytes ratio	12.4 [6.6–17.6]	10.1 [4.7–15.6]	14.1 [8.0–23.0]	0.01
Neutrophils/monocytes ratio	21.5 [14.8–34.7]	22.3 [13.2–35.0]	20.4 [15.8–25.7]	0.8
Neutrophils/platelets ratio	0.02 [0.02–0.03]	0.02 [0.02–0.03]	0.03 [0.02–0.04]	0.01
Fibrinogen (g/L)	6.4 [5.6–7.3]	6.2 [5.4–6.9]	6.9 [6.1–7.7]	0.03
D-Dimers (n = 80) (µg/L)	1128 [801–1914]	1022 [752–1407]	1341 [932–2679]	0.07
vWF: GP1b-binding activity (UI/dL)	280 [224–346]	268 [216–326]	286 [265–389]	0.03
vWF activity > 250 UI/dL (n, %)	55 (64.7%)	27 (54.0%)	28 (84.9%)	0.004
Prothrombin Fragment 1 + 2 (pM)	179 [134–323]	155 [129–248]	226 [151–578]	0.01
Prothrombin Fragment 1 + 2 > 500 pM (n, %)	14 (16.5%)	5 (9.6%)	9 (27.3%)	0.04
Prothrombin time ratio (%)	99 [90–100]	97 [89–100]	99 [90–100]	0.7
aPTT ratio	1.18 [0.98–1.35]	1.12 [1.01–1.27]	1.18 [0.98–1.36]	0.6

A-IPC: absolute immature platelet count; aPTT: activated prothrombin time; vWF: von Willebrand factor.

**Table 4 jcm-11-01383-t004:** Viscoelastic tests of COVID-19 patients and control patients.

Parameters	All Covid Patients (n = 85)	Non-Critical (n = 52)	Critical (n = 33)	“Control” Patients (n = 25)	*p* “Control” vs. Covid Patients	*p*-Value Non-Critical vs. Critical Covid Patients
CTH (s)	128 [113–137.5]	129 [112–140]	125 [116–133]	143 [132–147]	<0.0001	0.6
CS (hPa)	32.1 [24.1–25.5]	31.9 [24.3–22.5]	32.6 [22.5–40.6]	19.4 [17.6–25.5]	<0.0001	0.9
PCS (hPa)	24.5 [19.7–31.6]	24.5 [19.5–33.1]	24.5 [20.0–29.3]	17.3 [15.3–22.5]	0.0002	0.8
FCS (hPa)	7.4 [4.5–10.0]	6.6 [4.3–5.3]	8.4 [5.3–10.4]	2.1 [1.6–2.9]	<0.0001	0.2

CTH: clotting time with heparanase; CS: clot stiffness; PCS: platelet contribution to clot stiffness; FCS: fibrinogen contribution to clot stiffness.

**Table 5 jcm-11-01383-t005:** Associations between clinical and biological variables at ICU admission and worsening toward critical condition during the ICU stay.

Variable	Univariable Analysis		Logistic Regression Full Model ^a^		Logistic Regression Final Model ^b^	
	OR [95%CI]	*p*	OR [95%CI]	*p*	OR [95%CI]	*p*
Age (per 5 years)	1.56 [1.21–2.09]	0.001	1.23 [0.90–1.75]	0.2		
BMI (per kg·m^−2^)	0.97 [0.90–1.03]	0.3				
SAPS-II score	1.09 [1.04–1.14]	0.0003	**1.09 [1.04–1.17]**	**0.002**	**1.11 [1.06–1.17]**	**<0.0001**
Corticosteroid	0.20 [0.01–1.61]	0.2				
Anticoagulation	0.44 [0.08–2.15]	0.3				
A-IPC (per 5 G/L)	0.72 [0.51–0.96]	0.04	**0.46 [0.24–0.77]**	**0.009**	**0.47 [0.25–0.76]**	**0.007**
NLR (per 10%)	1.60 [1.08–2.49]	0.03	0.97 [0.52–1.80]	0.9		
NPR (per unit)	0.35 [0.01–60.18]	0.7				
Fibrinogen (per g/L)	1.34 [0.98–1.89]	0.08	1.45 [0.91–2.47]	0.1		
vWF activity (per 50 UI/dL)	1.29 [1.03–1.66]	0.04	1.08 [0.80–1.48]	0.6		
Prothrombin F1 + 2 (per 50 pM)	1.12 [1.02–1.24]	0.02	1.07 [0.94–1.22]	0.3		
D-Dimer (µg/L)	1.00 [0.99–1.00]	0.3				

Bold variables are included in the full model for logistic regression ^a^ and, if significant, in the final model ^b^ A-IPC: absolute immature platelet count; BMI: body mass index; NLR: neutrophils to lymphocytes ratio; NPR: neutrophils to platelets ratio; SAPS-II: Simplified Acute Physiology Score II; vWF: von Willebrand factor.

## Data Availability

The data that support the findings of this study are available from Caroline Thill at the biostatistics department, Rouen University Hospital, but restrictions apply to the availability of these data, which were used under license for the current study, and so are not publicly available. Data are however available from the authors upon reasonable request and with permission from Caroline Thill and the Research Department of the Rouen University Hospital (caroline.thill@chu-rouen.fr).

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
