# Peer review of "Routine and Advanced Laboratory Tests for Hemostasis Disorders in COVID-19 Patients: A Prospective Cohort Study"

_jcm, 2022, doi:10.3390/jcm11051383_

Round 1
Reviewer 1 Report
I would like to congratulate the authors on the results of the COVID 19 hemostasis cohort. This is an interesting research paper that contributes to an important topic of coagulation disorders in COVID-19 patients. In general the article is well written, concise ,but there are several issues that have to be addressed and resolved:
- Title of the manuscript: Routine and Advanced laboratory tests for hemostasis disorders during in Covid-19 patients: a prospective cohort study – I would suggest omitting „during“ and use capital letters for „COVID-19“ as usually throughout the whole article
- Abstract: Background: Use „Hence“ instead of „Because“, Materials: instead of „worsening“ use „worse outcome“
- Line 157 (cf supra) – what does it mean?
- Line 185 – what was the reason for „wrong“ COVID pneumonitis diagnosis? Were the patient tested PCR negative or different reason. You have to be more eleborate in explaining, why the patients were excluded. Exclusion of 13% of patients for technical issues with laboratory diagnostics is considerably high number.
- Table 1. expressing the continuous values as mean± standard deviation is preferable. Tables should be self-explanable , thus explain all the abbreviations inlcuding SAPS and SOFA score, NIV,HFNC
- Table 2. Again labelling as worsening vs. non-worsening sounds somewht weird, i would suggest using stable (or non-critical) and critical condition.
- Figure 1. Also I do find easier to use acronyms for worsening/non-worsening (critical/non-critical) instead of the whole wording. NC vs. C with explanation in the figure description
- Table 5 – the statistics behind the full and final model is not clear to me. Maybe the explnation for the full model disappeared from my version of the manuscript while there is no small „a“ under the table. If you claim that A-IPC is independently associated with development of critical condition, did the model entered some co-variables as antropomethric parameters, co-morbidities, antiviral agents used or plain platelet count etc.? In other words did you look for confounder?
- Line 295 as discussed not discuss
- Line 369-370 – this sentence is hard to read, please rewrite. You wanted to say that the new dye is more specific for mitochondrial and platelet cytosolic RNA?
- Line 373 the biomarker is one (A-IPC), so use singular
Author Response
We would like to thank the reviewer for her/his relevant comments, allowing to improve the manuscript. Please find below a point-to-point response.
Title of the manuscript: Routine and Advanced laboratory tests for hemostasis disorders during in Covid-19 patients: a prospective cohort study – I would suggest omitting „during“ and use capital letters for „COVID-19“ as usually throughout the whole article
The title has been corrected and COVID-19 has been changed throughout the manuscript
Abstract: Background: Use „Hence“ instead of „Because“, Materials: instead of „worsening“ use „worse outcome“
It has been corrected
Line 157 (cf supra) – what does it mean?
In this section, we described the objective of the study, but we agree that it was somewhat unclear. We modified the sentence to clarify the purpose (line 205-207)
Line 185 – what was the reason for „wrong“ COVID pneumonitis diagnosis? Were the patient tested PCR negative or different reason. You have to be more eleborate in explaining, why the patients were excluded. Exclusion of 13% of patients for technical issues with laboratory diagnostics is considerably high number.
Concerning the false covid pneumonitis diagnosis, a patient presented a respiratory failure related to an interstitial pneumopathy and another with a false positivity of PCR (not confirmed by later testing). Concerning the technical issues, 5 presented early coagulation within the sample tube not suitable for analysis and 8 presented an absence of clotting throughout the analyses. We decided to also excluded these patients because of the very likely technical issues, which may have altered the results (these patients were not under important anticoagulant therapy). We added these precisions lines 239-243.
Table 1. expressing the continuous values as mean± standard deviation is preferable. Tables should be self-explanable , thus explain all the abbreviations inlcuding SAPS and SOFA score, NIV,HFNC
We decided to express the results as medians with interquartile ranges because of the absence of normality of the data. Thus, the mean may be not representative of the distribution of data and. Moreover, comparisons were realized using a Mann-Whitney test because of the absence of a normal distribution of data, thus observed significance (or the contrary) may not be reflected by representation of means. It seems to us more precise to present data as medians. We added a sentence in the “statistical analyses” section to justify this choice.
Table 2. Again labelling as worsening vs. non-worsening sounds somewht weird, i would suggest using stable (or non-critical) and critical condition.
We modify the groups as “non-critical” and “critical” throughout the manuscript.
Figure 1. Also I do find easier to use acronyms for worsening/non-worsening (critical/non-critical) instead of the whole wording. NC vs. C with explanation in the figure description.
It has been modified
Table 5 – the statistics behind the full and final model is not clear to me. Maybe the explnation for the full model disappeared from my version of the manuscript while there is no small „a“ under the table. If you claim that A-IPC is independently associated with development of critical condition, did the model entered some co-variables as antropomethric parameters, co-morbidities, antiviral agents used or plain platelet count etc.? In other words did you look for confounder?
We realized a logistic regression model including variables using a backward stepwise process among: BMI, age, SAPS-II severity score, covid-19 specific treatment: corticosteroid, anticoagulant, and biologic results. Only variables with a significant association with worsening in univariable analysis were included. The final model was a repetition of the initial model but without the non-significant variables (in the multivariable analysis) to obtain more precise OR. It is now described in the “statistical analysis” section
Line 295 as discussed not discuss
It has been corrected
Line 369-370 – this sentence is hard to read, please rewrite. You wanted to say that the new dye is more specific for mitochondrial and platelet cytosolic RNA?
The sentence has been simplified
Line 373 the biomarker is one (A-IPC), so use singular
It has been corrected
Reviewer 2 Report
I have attached notes to the manuscript below:
1. The reference for the first sentence in the introduction should be given.
2. L46 - this sentence lacks an essential systematic review on viscoelastic tests in COVID-19 (DOI: 10.1055/a-1346-3178.).
3. Criterion should be written in plural?
4. In addition, the materials and methods are clearly described, the authors correctly defined the clinical conditions associated with COVID-19.
5. In the description of laboratory tests, authors should provide reference values (perhaps in supplementary materials).
6. The results are precise. Please improve the quality of the figures. The discussion is interesting.
Overall, well written and exciting work. After considering my comments, I recommend the paper be published in JCM.
Author Response
We would like to thank the reviewer for her/his very kind and relevant comments on our manuscript. Please find below a point-to-point response
The reference for the first sentence in the introduction should be given.
We added two references, one from the covid-19 daily dashboard of WHO and a study from Gavriatopoulou et al describing the main organ failures related to coid-19 infection (ref 1 and 2)
L46 - this sentence lacks an essential systematic review on viscoelastic tests in COVID-19 (DOI: 10.1055/a-1346-3178.).
We added this reference ([8])
Criterion should be written in plural?
It has been corrected
In addition, the materials and methods are clearly described, the authors correctly defined the clinical conditions associated with COVID-19.
Thank you for your comment
In the description of laboratory tests, authors should provide reference values (perhaps in supplementary materials).
We added reference values in the supplementary file 1
The results are precise. Please improve the quality of the figures. The discussion is interesting.
Thank you for your kind remark. The quality of the picture may have been reduced by importation in the manuscript. We provide high quality picture, realized under GraphPad Prism, to the editor for publication (600 dpi).
Round 2
Reviewer 2 Report
The authors have satisfactorily addressed all of my concerns.